

# EarthCARE's Cloud Profiling Radar Antenna Pointing Correction using Surface Doppler Measurements

Bernat Puigdomènech Treserras[1], Pavlos Kollias[1,2], Alessandro Battaglia[3,4], Simone Tanelli[5] and Hirotaka Nakatsuka[6]

[1]Department of Atmospheric and Oceanic Science, McGill University, Montreal, H3A 0B9, QC Canada

[2]School of Marine and Atmospheric Sciences, Stony Brook University, Stony Brook, NY 11790, NY USA

[3]Department of Environment, Land and Infrastructure Engineering (DIATI), Politecnico of Torino, Turin, Italy

[4]Department of Physics and Astronomy, University of Leicester, Leicester, UK

[5]Jet Propulsion Laboratory, California Institute of Technology, Pasadena CA USA

[6]Japan Aerospace Exploration Agency, 305-8505 2 Chome–1–1, Sengen, Tsukuba, Ibaraki, Japan

*Correspondence to*: Bernat Puigdomènech Treserras (bernat.puigdomenech-treserras@mcgill.ca)

**Abstract.** The Earth Cloud Aerosol and Radiation Explorer (EarthCARE) mission, a joint effort between the European
Space Agency (ESA) and the Japan Aerospace Exploration Agency (JAXA), aims to advance our understanding of aerosols, clouds, precipitation, and radiation using a comprehensive active and passive sensors payload. A key component of the payload is the 94-GHz Cloud Profiling Radar (CPR), which provides the first-ever Doppler velocity measurements collected from space. Accurate knowledge of the CPR antenna pointing is essential for ensuring high quality CPR Doppler velocity measurements. This study focuses on the geolocation assessment and antenna mispointing corrections during EarthCARE's
commissioning phase and beyond, using Earth's surface Doppler velocity measurements collected over the first nine months of the mission. While the instrument footprint is proven to be properly geolocated within about 100 meters, surface Doppler velocity observations reveal mispointing trends influenced by solar illumination cycles and thermoelastic distortions on the antenna. Correcting these effects significantly reduces biases, ensuring better Doppler velocity measurements, essential for understanding cloud microphysics and dynamics. The results, validated through the analysis of Doppler velocities in ice
clouds, underline the critical role of pointing corrections for the success of the EarthCARE mission.



## 1 Introduction

The Earth Cloud Aerosol and Radiation Explorer (EarthCARE) mission (Wehr et al., 2023), a joint satellite mission by the European Space Agency (ESA) and the Japan Aerospace Exploration Agency (JAXA), was successfully launched on May 28th, 2024. The mission aims to deliver groundbreaking observations of aerosols, clouds, precipitation, and radiation, advancing our understanding of their properties and intricate interactions that will help improve climate models and weather forecasting (Illingworth et al., 2015). As the most sophisticated ESA's Earth Explorer mission to date, the EarthCARE satellite is equipped with two active (radar and lidar) and two passive (spectral and broadband) radiometer sensors. The on-board radar is a high-sensitivity 94 GHz Cloud Profiling Radar (CPR) with Doppler capabilities, enabling the first-ever collection of Doppler velocity measurements from a spaceborne radar system (Kollias et al., 2014; Battaglia et al., 2020).

During the commissioning phase that included the first six months following its launch, several activities were performed to finalize and ensure the proper operation of the satellite's payloads according to the mission requirements. These activities include ground segment checks for data acquisition, processing, and distribution, as well as the verification of health and functionality, in-orbit calibration, characterization, and performance verification of the instruments. The experience gained by the successful CloudSat mission by the National Aeronautics and Space Administration (NASA), the Canadian Space Agency (CSA) and the US Air Force (USAF) (Stephens et al., 2002) and its Cloud Profiling Radar (Tanelli et al., 2008) provided a wealth of information on how to address several aspects of the initial CPR evaluation during the commissioning phase. This includes methods to calibrate CPR science data using the ocean surface normalized radar cross section (Tanelli et al., 2008; Battaglia and Kollias, 2015a) and the CloudSat-derived climatology of ice clouds (Battaglia and Kollias, 2015b); ground clutter removal (Burns et al., 2016) and the Path Integration Attenuation (PIA) estimation (Haynes et al, 2009).

On the other hand, the EarthCARE CPR Doppler velocity measurements are new. The quality of these measurements is affected by three main factors (Kollias et al., 2022): intrinsic noise due to the signal decorrelation (spectral broadening) introduced by the platform motion, residual errors in correcting Doppler velocity biases introduced by non-uniform beam filling (Tanelli et al., 2002; Kollias et al., 2014; Sy et al., 2014) and outstanding biases and errors due to uncertainty in the CPR antenna pointing characterization (Tanelli et al., 2005;  Battaglia and Kollias, 2015b). The treatment of the two first terms (spectral broadening and NUBF) in the CPR L2a data products is described in Kollias et al., 2023. Here, we focus on two critical activities related to the third term, conducted during the commissioning phase and extending a few months beyond: the geolocation and the assessment of the off-nadir antenna pointing angle along the orbital track, especially important to determine the quality of the Doppler velocity measurements. These activities are based on surface measurements collected over natural targets between the months of August 2024 and February 2025, the geolocation techniques described in Puigdomènech Treserras and Kollias, 2024 and the C-APC product described in Kollias et al., 2023. The results presented here are further validated through a comparison of the pointing effects on the climatology of Doppler velocities in ice clouds.



## 2 Geolocation

The accurate determination of the precise location on Earth's surface and atmosphere corresponding to signals received by the CPR instrument is essential for their interpretation. Furthermore, since one of the strengths of the EarthCARE mission is the synergistic use of multisensor observations, the CPR measurements must be properly geolocated in order to ensure an effective integration with the signals from all other sensors. These measurements are combined in the synergistic algorithms, like AC-TC (Irbah et al., 2023), ACM-CAP (Mason et al., 2023) and ACM-COM (Cole et al., 2023).

Here, the geolocation assessment is performed using the techniques described in Puigdomènech Treserras and Kollias 2024, based on the positions of known natural targets, such as significant elevation gradients and coastlines over more than 140 domains of 2x2 degrees distributed around the globe. For significant elevation gradients, the assessment is performed by comparing the instrument's surface detection height to a reference digital elevation model. Artificial mispointing errors are introduced in the along- and cross-track directions, and the absolute geolocation is determined by the shift that maximizes

the correlation between the instrument and the DEM-estimated surface height. For coastlines, the instrument's surface signal gradients between land and ocean transitions are leveraged to model the coastline signatures. Then, through a minimization approach, the absolute geolocation is determined by minimizing the error between a collection of coastline detections and a reference map.

During the commissioning phase, an extensive analysis of the data collected over the specified set of regions of interest

confirmed that the CPR instrument is accurately geolocated, with overall pointing errors remaining below 100 m—five times better than the initially specified requirements by ESA.

Figure 1 illustrates two examples of the geolocation analysis over a mountainous region of British Columbia, Canada, and the southern coastlines of Greece. The first analysis, illustrated in panels (a) and (b), utilizes the significant elevation gradients technique based on a single overpass, which has proven sufficient for performing the geolocation assessment. This

technique benefits from the abundance of surface detections within the domain given by the along-track resolution of 500 meters and the vertical sampling resolution of 100 meters. In contrast, the coastline geolocation assessment, illustrated in panel (c), is derived from a collection of detections spanning four months of data used in the analysis, August to November 2024. This approach is necessary because the coastline technique typically identifies only one or a few crossings within the domain that is not enough to assess the instruments' geolocation.

Overall statistics of the geolocation assessment are presented in Figure 2, which shows that the average along- and cross-track mispointing angles are 0.002º and 0.008º for ascending orbits and –0.014º and –0.007º for descending orbits, respectively. At a satellite altitude of 395 km, a mispointing error of 0.01º corresponds to a geolocation error of approximately 69 m. While the geolocation techniques have inherent accuracy limits, as described in Puigdomènech Treserras and Kollias (2024), the results show the presence of an along- and cross-track biases that are well within the

pointing requirements.





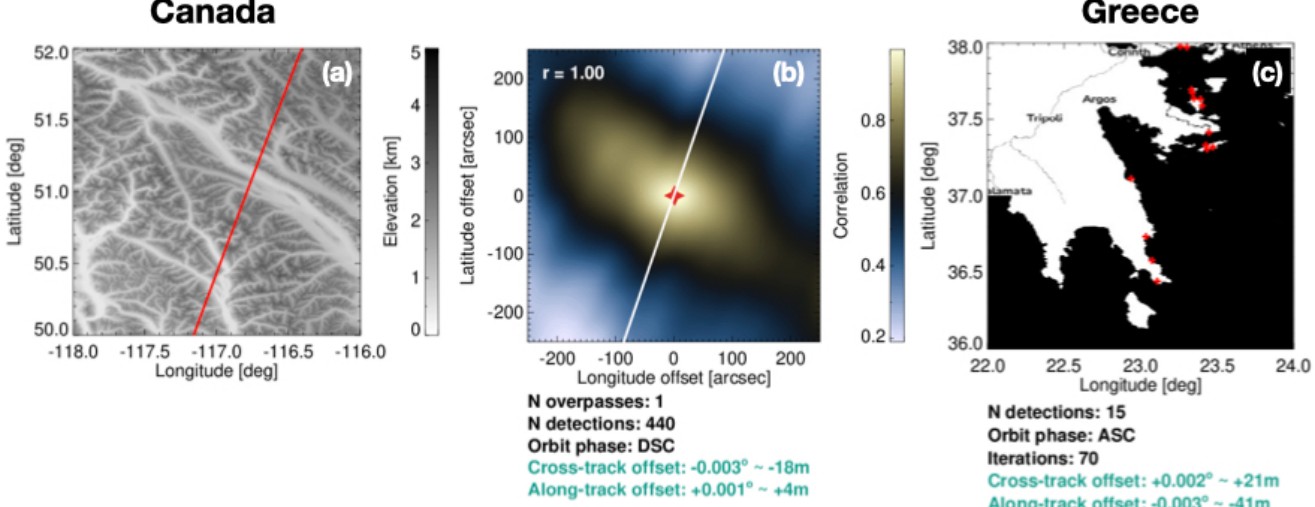

**Figure 1: EarthCARE CPR geolocation assessment using significant elevation gradients and coastlines. Panel (a) illustrates a mountainous region in British Columbia, Canada, with red line representing one of the EarthCARE overpasses. The statistical correlation analysis is depicted in panel (b), with the white line representing the satellite path, in descending orbit, and the red filled star denoting the final geolocation offset. Panel (c) shows the geolocation assessment using Greek coastlines, with the red dots representing clear detections collected from August to November 2024. The base map is © OpenStreetMap contributors 2015, distributed under the Open Data Commons Open Database License (ODbL) v1.0.**

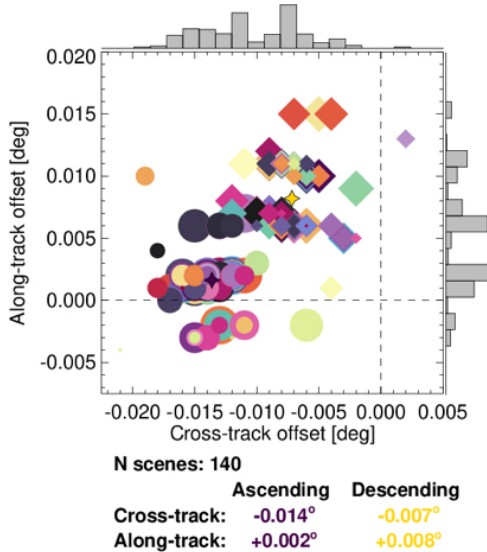

**Figure 2: Combined global geolocation statistics of the EarthCARE CPR for data collected from August to November 2024. Each symbol represents an individual domain where the geolocation is assessed, with the symbol size being indicative of the number of overpasses. Circles and diamonds correspond to ascending and descending parts of the orbit, respectively. Distinctive colors identify different domains, while filled stars denote the averages: dark purple for ascending and yellow for descending. The dashed lines denote the perfect geolocation point (0º).**



## 3 Surface Doppler velocity

Although the CPR instrument footprint is geolocated within the specified requirements (~ 1/10 of its footprint), this requirement is not sufficient to ensure that no Doppler velocity biases are introduced due to antenna mispointing. At the velocity of the satellite, about 7.6 km/s, a minimal mispointing error of 0.01° in the along-track direction translates into a

Doppler velocity bias of about 1.33 m/s. Therefore, meeting the geolocation requirements in not satisfactory for achieving good Doppler measurements. Thus, a more detailed assessment is required to identify and possibly correct any possible Doppler velocity bias due to along-track antenna mispointing. To evaluate this further, the Doppler velocity measurements are analyzed to verify if there is any residual mispointing contaminating the data.

Fortunately, the Earth's surface, that typically represent a disturbance for the atmospheric signal, can be used as a Doppler

signal reference target. For a Doppler radar pointing near nadir, when a beam-limited approximation is valid, assuming that the antenna footprint is several 100s of m wide, it can be assumed that the average vertical velocity of the surface (be it ocean, vegetated land or other) is generally zero in average. Therefore any departure from the expected 0 m/s velocity indicates a potential antenna mispointing (Testud et al., 1995; Kobayashi and Kumagai, 2003; Tanelli et al. 2005; Battaglia and Kollias, 2015a; Scarsi et al., 2024).

The analysis of surface Doppler velocities is performed on individual orbits to ensure both temporal and spatial consistency in the data. For each orbit, surface measurements are examined within 250 km along-track running windows, calculating the averages and standard deviations for each window. To provide a robust summary of the data, the median of these statistics is computed for each position globally, gridded at a resolution of 1°x1° latitude and longitude degrees.

This method is chosen for several reasons. First, using individual segments allows for a clear separation of measurements in

time and space, reducing potential biases from overlapping data. After examining the data at different window lengths, the 250 km window (i.e. about 32 s integration time) is chosen as an optimal balance: it is long enough to smooth out small-scale variability, such as noise, while still preserving meaningful large-scale trends in the Doppler measurements. Finally, employing the median of the averages and standard deviations effectively minimizes the impact of outliers, providing a reliable representation of the overall Doppler velocity average and variability, while maintaining robustness to noise and

anomalies.

The spatial average and variability of the surface Doppler velocity (i.e. the Doppler velocity corresponding to the surface range) for the period from August to November 2024 is shown in Figure 3a,b. The corresponding sea ice coverage and snow-covered land areas derived from the ECMWF model are shown in Figure 3c. Variance in the measured surface Doppler velocity arises not only from surface type but also from spectral broadening, which results from the convolution of the CPR

antenna pattern with the velocity gradients within the radar footprint due to the satellite's rapid motion (Sy and Tanelli, 2022). Additionally, the CPR pulse-repetition frequency (PRF) configuration, shown in Figure 3d, determines the number of pulses transmitted per second by the CPR and thus the number of samples used to estimate the surface Doppler velocity, affecting its uncertainty. The higher the PRF value, the lower the Doppler velocity uncertainty (Kollias et al., 2014). The



depth of the atmospheric layer we want to sample (troposphere) is the only practical limiting factor. As a result, the PRF

progressively increases at higher latitudes as the depth of the troposphere decreases.

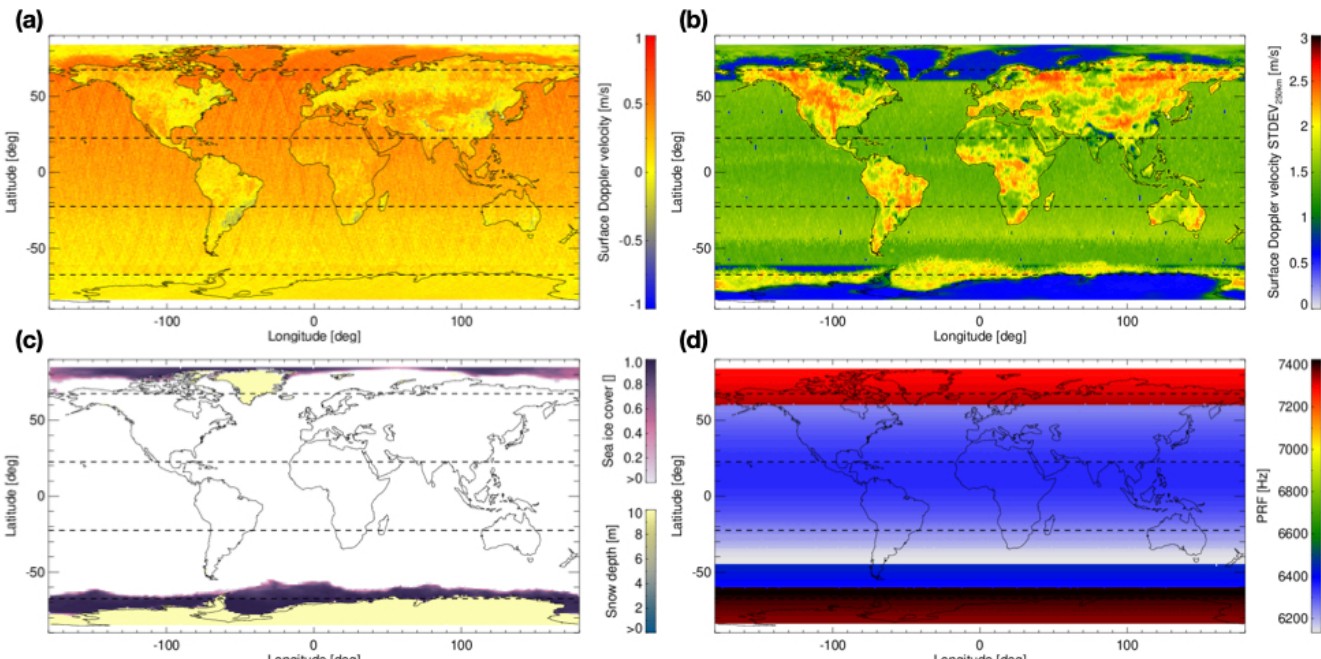

**Figure 3: Panels (a) and (b) depict the global surface Doppler velocity tendency and variability, represented by the median of the mean and standard deviation values, both calculated from orbit-to-orbit measurements within a 250 km along-track window**
**collected from August to November 2024. Panel (c) depicts the ECMWF average model sea ice cover and snow depth for the same period, and panel (d) the default variable EarthCARE CPR's pulse-repetition frequency (PRF) configuration changing as a function of latitude. The horizontal dashed lines are plotted to indicate the boundaries of the orbit's segments, defined by frame IDs: (A) +22.5° to -22.5° ascending, (B) +22.5° to +67.5° ascending, (C) +67.5° ascending to +67.5° descending, (D) +67.5° to +22.5° descending, (E) +22.5° to -22.5° descending, (F) -22.5° to -67.5° descending, (G) -67.5° descending to -67.5° ascending, and (H) -**
**67.5° to -22.5° ascending.**

While the results highlighted in Figure 3 do not differentiate between ascending and descending orbits, panel (a) already reveals a noticeable bias in surface Doppler measurements as a function of latitude, suggesting potential mispointing, especially in the northern hemisphere (darker colors). At least two primary factors could contribute to this mispointing: errors in the satellite's attitude systems and thermoelastic distortions. Panel (b) depicts the surface Doppler variability, panel

(c) depicts the ECMWF average model sea ice cover and snow depth and panel (d) the default EarthCARE CPR's PRF configuration.

One of the most notable characteristics of the surface Doppler measurements is their variability, which clearly correlates with surface type, orography, and the CPR PRF settings. The lowest Earth's surface Doppler velocity variability is observed over ocean and snow-covered land (e.g., Antarctica and Greenland). Flat surfaces are expected to introduce no vertical

motion at nadir, whereas rough topography can generate significant terrain-induced Doppler effects due to slopes and



variations in reflectivity causing non-uniform beam filling effects (Manconi et al., 2024). Consequently, land regions tend to exhibit noisier measurements, with exceptions such as the deserts of Western Australia, the Sahara, and Namibia, which have relatively uniform surfaces. Sea ice, on the other hand, appears to considerably increase the measurement variability. Additionally, the high PRF settings, configured to find balance between the unambiguous range and the tropopause height (a

proxy for maximum cloud top height) at different latitudes, significantly reduce the measurement variability at high latitudes (e.g., near the North Pole and Antarctica) where the PRF is at its high, further highlighting the influence of the instrument configuration on data quality.

## 4 Antenna Pointing Correction

The clear-sky Doppler velocity measurements over the ocean (free of ice) and snow-covered land (Antarctica and

Greenland) collected for all orbits from June 2024 to February 2025 are used to document the biases observed in the global climatological analysis and, in order to identify any potential antenna mispointing, The surface Doppler velocity measurements are corrected for non-uniform beam filling effects, averaged using a 250 km running window and converted into antenna mispointing angles considering the satellite velocity. Figure 4 illustrates the resulting weekly averaged mispointing angles as a function of ANX time (time since ascending node crossing).

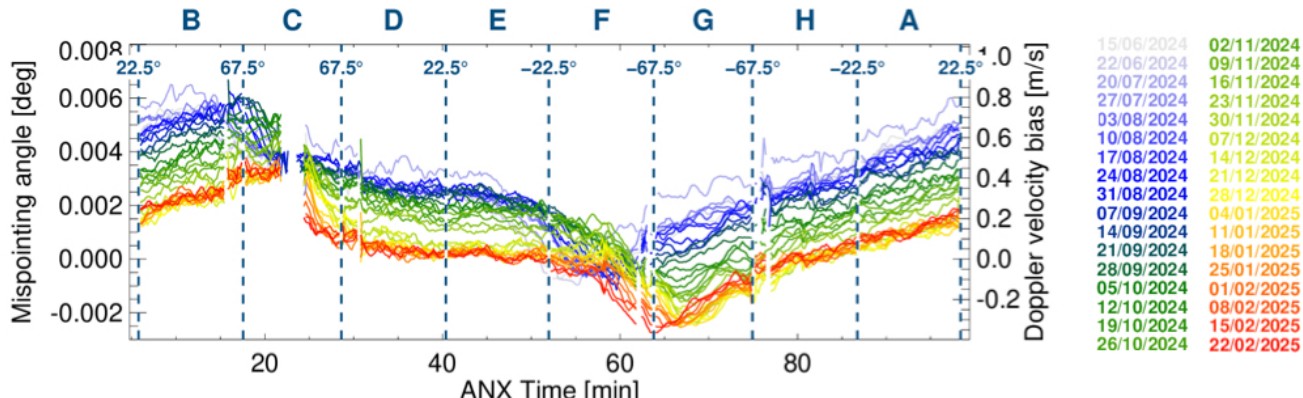


**Figure 4: Weekly averaged EarthCARE´s CPR antenna mispointing angle as a function of ANX time (time since ascending node crossing) derived from clear-sky surface Doppler velocity measurements collected over sea surface (free of ice) and snow-covered land from June 2024 to February 2025. The letters on top correspond to the frame ID, which denote the different segments of the orbit, each spanning a specific latitude and time range marked by the dashed vertical lines, with the corresponding latitude values**
**displayed above them.**

The analysis presented in Figure 4 demonstrates a repeating pattern along the orbit as a function of the ANX time; the antenna mispointing increases reaching a peak and then it starts to gradually decline until reaching a minimum, before it begins to increase again, ultimately reaching its maximum peak in the next orbital cycle. Applying a 250 km running window and performing a weekly average effectively smooths the data, revealing the underlying trend with an amplitude of

approximately 0.006º (0.8 m/s). Both the phase and amplitude of this trend shift over the course of the year. In June, the



mispointing peak reaches 0.006º before the 20-minute mark, while the minimum is around 0º at the 55-minute mark. In February, the mispointing trend remains similar but appears shifted by nearly 10 minutes in time, with the amplitude's maximum and minimum reaching approximately 0.004º and -0.002º, respectively. This amplitude span of 0.006º is significant because corresponds to a Doppler velocity shift of 0.8 m/s. The deviation of individual 250 km along-track

averaged measurements from their respective fits exhibits a second-order residual with a standard deviation of approximately 0.00055º (0.07 m/s). Additionally, individual orbits occasionally diverge from their weekly averaged trend, likely due to variations in solar radiation (Bard and Frank, 2006) affecting the antenna deformation pattern. Since the presence of thermoelastic distortions cannot be excluded, the effect of sunlight rays reaching the antenna must be considered. Specifically, the spacecraft daylight entry and exit times, along with the solar azimuth angle and their seasonal variations that

are expected to cause the shifts in time and amplitude.

Another important point worth noting is the similarity with the along-track component of the previously introduced geolocation assessment. Both analyses yield results of the same order of magnitude, with differences arising from methodological differences between the two approaches.

To fully characterize the evolution of the CPR antenna mispointing throughout the year, the minimum and maximum values

of each weekly averaged mispointing trends are tracked and analyzed as a function of time of year. The results are highlighted in Figure 5.

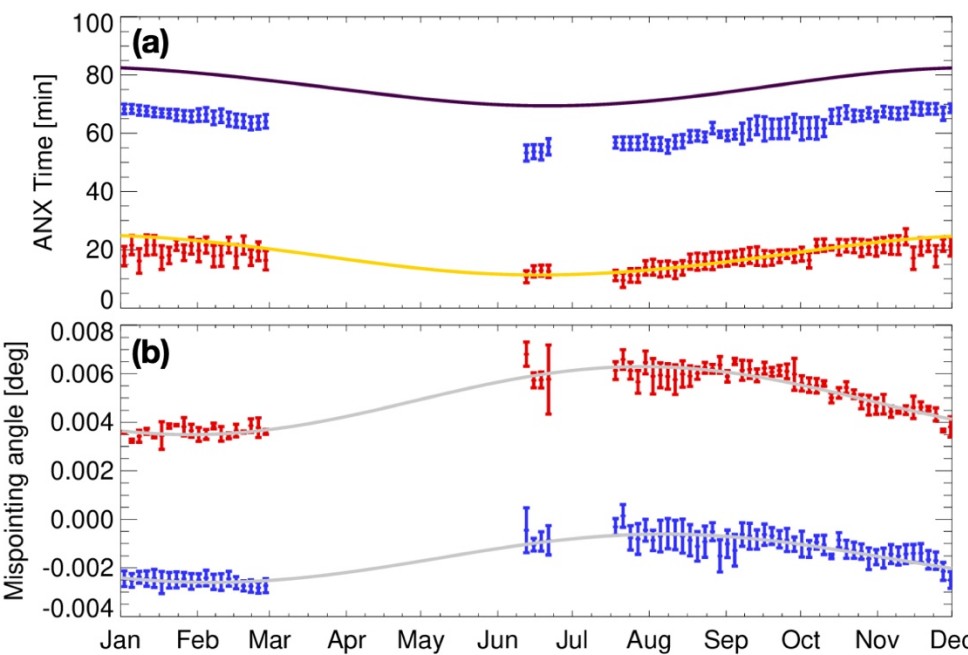

**Figure 5: Panel (a) shows the ANX time of the minimum (blue) and maximum angles (red) of the weekly averaged mispointing**
**trends as a function of time of year. The yellow and purple lines denote the spacecraft's daylight entry and exit times, respectively.**
**Panel (b) presents the minimum (blue) and maximum (red) angles of the weekly averaged mispointing trends as a function of time**



of year, with grey lines representing fitted harmonics. In both panels, the vertical bars indicate the standard deviation. Gaps correspond to periods not yet covered by CPR measurements.

These results illustrate a dependence between solar illumination and the mispointing cycle, supporting the hypothesis that thermoelastic distortions affect the CPR antenna pointing. The mispointing trend reaches its maximum when the spacecraft enters daylight and its minimum about 12 minutes before exiting daylight. The observed increase in mispointing in Figure 4 corresponds to the eclipse phase of the orbit, while the gradual decline occurs during the sunlit phase. Additionally, the mispointing amplitude varies throughout the year. During winter in the Northern Hemisphere, both the minimum and maximum mispointing values are at their lowest, whereas their magnitudes increase during the summer months. This analysis indicates that both the time and amplitude shifts observed in Figure 4 can be explained by seasonal variations, specifically the changes in solar elevation—affecting the spacecraft's entry and exit times—and solar azimuth, which influences the incidence of sunlight on the antenna. These variations evolve systematically with ANX time throughout the year and thus are fully predictable.

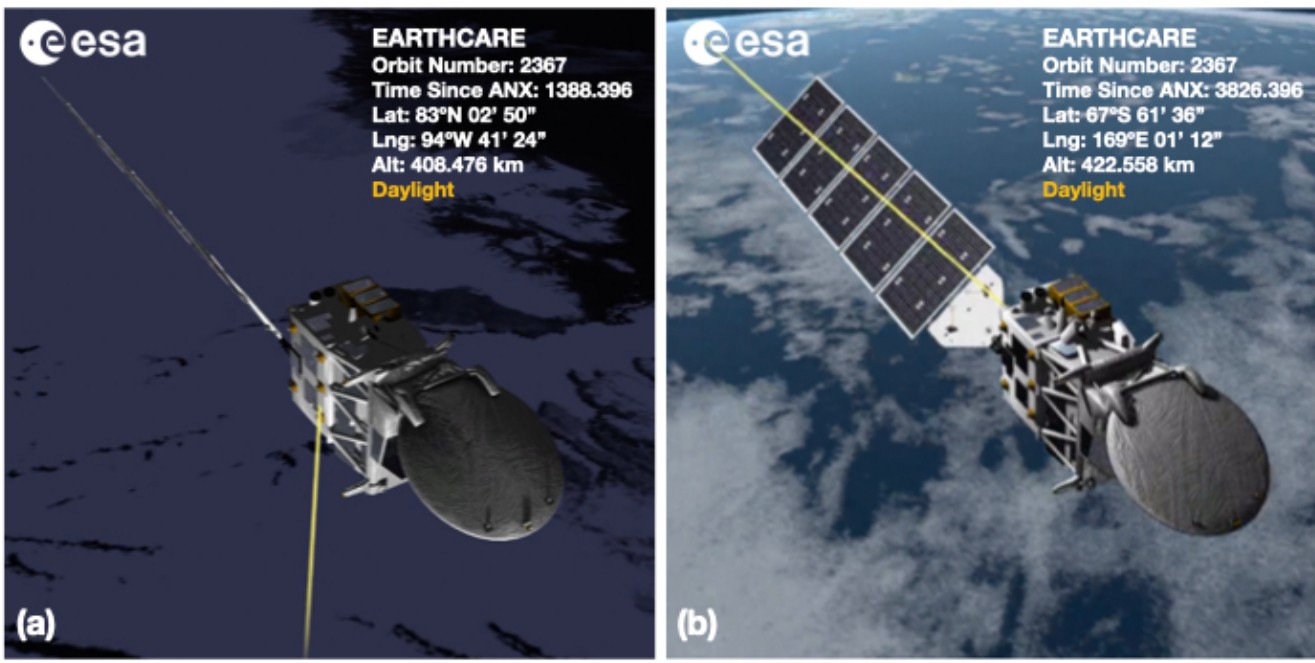

**Figure 6: 3D model of EarthCARE from the Satellite Mission Editor & Player (SAMI) software package distributed by ESA. Panel (a) shows the satellite a few seconds after entering daylight, while panel (b) depicts it a few minutes before exiting daylight on October 1st, 2024. The yellow line represents the direction of sunlight rays reaching the satellite.**

Figure 6 provides a visualization of EarthCARE's orientation relative to sunlight during its daylight phase, helping explain the antenna mispointing trends. Panel (a) shows the spacecraft just a few moments after entering daylight, where sunlight rays (yellow line) strike the antenna perpendicularly. This direct solar illumination likely causes the rapid change in the mispointing angle trend observed early in the daylight phase (frames B and C in Figure 4). Panel (b) illustrates the spacecraft shortly before exiting daylight. At this point, sunlight is partially blocked by the spacecraft body and solar panels, reducing



direct exposure to the antenna. This shading effect seems to contribute to the second change in trend, as observed after the decline toward near-zero values (frame F and G in Figure 4). The transition between these two phases—direct exposure upon entry and gradual shading before exit—aligns with the periodic variations in the mispointing angle and highlights the impact of prolonged solar illumination and its absence, consistent with thermoelastic distortions. The delays or offsets in the change of trends are likely a result of the time required for thermal effects to propagate through the antenna structure.

The information presented in Figure 5 is used to establish a normalized parametrization of the antenna mispointing pattern in Lagrangian coordinates, with January 1st as the reference time. This parametrization accounts for the systematic seasonal variation that affects both phase and amplitude shifts. The resulting fit, shown in Figure 7, provides a refined characterization of the climatological antenna mispointing pattern.

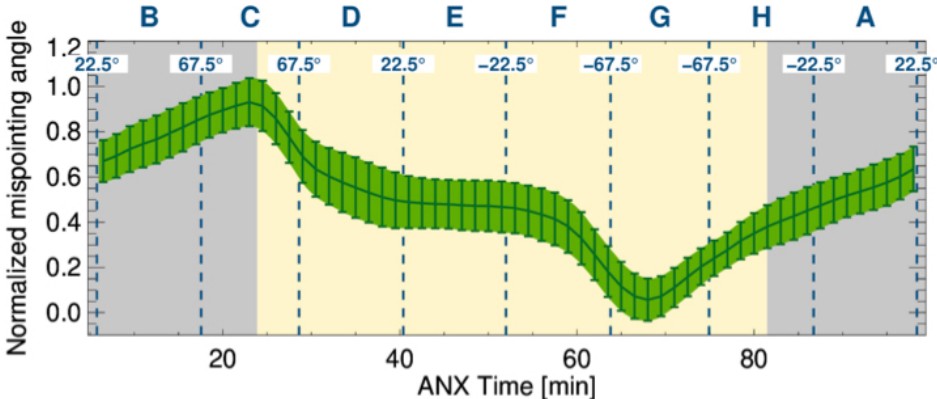

**Figure 7: Normalized antenna mispointing angle as a function of the ANX time, aligned with the spacecraft's daily daylight cycle as of January 1st. The angle is normalized according to the seasonal amplitude variation. The green lines and shading denoting the mean and standard deviation, and the yellow background represents the spacecraft daylight coverage time. The letters on top correspond to the frame ID, which denote the different segments of the orbit, each spanning a specific latitude and time range marked by the dashed vertical lines, with the corresponding latitude values displayed above them.**

This parametrization serves as a climatological reference for modeling the CPR antenna mispointing and can be used to identify the actual mispointing on an orbit-to-orbit basis. The phase and amplitude of the normalized fit are adjusted throughout the year, leveraging the information depicted in Figure 5. To determine the actual mispointing angle at any given time, the normalized mispointing value is first shifted based on the difference of the daylight entry time relative to the reference date (January 1st) and then scaled according to the seasonal amplitude variation. This ensures that the final estimate accurately reflects the expected mispointing trend over the year.

It is also worth noting that the previously mentioned second-order residual of 0.00055º (1.98 arc seconds), observed in the individual weekly orbital assessments, is also reflected in the variability of the normalized trend. Specifically, the standard deviation of the green shading is 0.9, which, when multiplied by the average amplitude of 0.006º (Figure 5b), results in 0.00055º, which corresponds to 7 cm/s in the Doppler velocity space. The convergence of these values suggests that the applied transformations accurately preserve the structure of the mispointing variability.



When applying this climatological reference to correct the EarthCARE CPR dataset, it is also essential to account for other
sources of uncertainty and orbit-to-orbit variations. As previously mentioned, some orbits occasionally diverge from the
expected trend suggesting that the antenna does not always deform in the exact same way. To mitigate residual biases and
ensure a highly adaptive and robust correction methodology, a final optimization step minimizes the residuals relative to the
mispointing angles derived from the ingested 250 km along-track averaged surface Doppler velocity observations.

This procedure aims to further adjust and reduce the residuals. The amplitude limits are systematically perturbed in small
increments of 0.0001° over a range spanning twice the measured standard deviation (0.0011°). For each perturbation pair, the
mean absolute difference (MAD) between the reference and the orbital observations is computed. After evaluating all
combinations, the optimal amplitude limits are determined by selecting the pair that minimizes the MAD. To evaluate its
effectiveness, this process has been applied to approximately 3,000 orbits. A comparison between the modeled mispointing
trend and the mispointing angles derived from the ingested observations shows that the 90th percentile of residuals remains
below 0.00077° (2.77 arc seconds, ~10 cm/s), highlighting the effectiveness and precision of the suggested method for
correcting the antenna mispointing.

## 5 Effect on Ice Clouds

As a further evaluation, the effects of CPR antenna mispointing and the proposed correction methodology are analyzed on
Doppler velocity measurements of ice clouds. For this purpose, the quality-controlled mean Doppler velocity estimates and
radar reflectivity of ice clouds from the C-CD and C-FMR product (Kollias et al., 2023) are collected from one of the time
periods where the antenna mispointing is at its maximum — January 2025. The C-CD processing includes a correction for
non-uniform beam-filling effects, integration over 4 km along-track and 500 m in height, and a correction for velocity
folding, while the C-FMR processing applies a filtering mask for non-meteorological echoes and a correction for gaseous
attenuation. Leveraging this information, the evaluation is performed using the global reflectivity–Doppler velocity (Z–V)
relationship of ice clouds between the temperatures of -30 to -40 degrees. The results of the analysis are depicted in Figure 8.

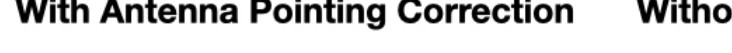

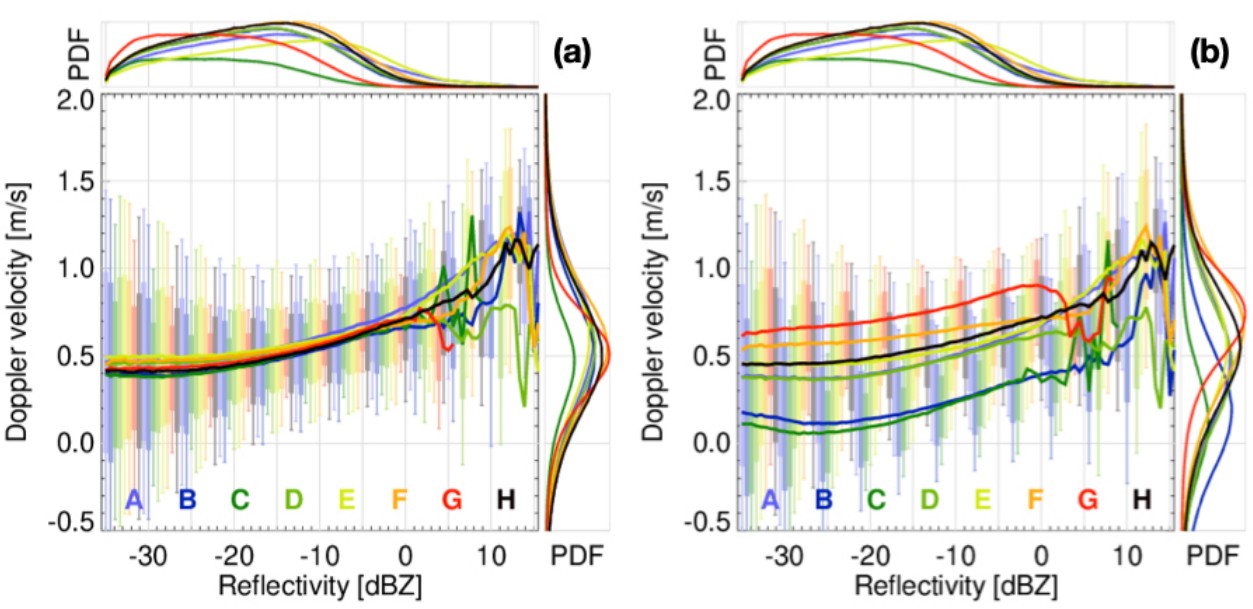

**Figure 8: Reflectivity–Doppler velocity (Z–V) relationships of ice clouds for temperatures between -30°C and -40°C for the month of January 2025. Panels (a) and (b) depict the Z–V relationships with and without the antenna pointing correction, respectively. Each color represents a different frame ID, while the vertical bars indicate the 10th, 25th, 75th, and 90th percentiles, and the solid lines represent the median. The overlaid probability density functions (PDF) on the top and right axes illustrate the distribution of reflectivity and Doppler velocity number of samples contributing to each range. Note that the C-CD L2a product inverts the velocity sign with respect to the C-NOM L1b product**

The analysis depicted in Figure 8 confirms the effectiveness of the antenna pointing correction. With the correction applied, the Z–V relationships become consistent across all different segments of the orbit, identified by the unique frame IDs. In contrast, without the correction, the Doppler velocities exhibit segment-dependent biases, with frames G and C being the most affected by positive and negative biases, respectively. This result agrees with the initial assessment shown in Figure 4. Additionally, the figure also illustrates the Doppler velocity variability under different signal-to-noise ratio (SNR) conditions. Below -21dBZ (SNR = 0), the Doppler velocity variability becomes significantly large, indicating a reduced measurement reliability.

Another important aspect worth mentioning is that the well-characterized Z–V climatological relationships of ice clouds from the EarthCARE dataset can serve as an additional reference for assessing the antenna mispointing (Battaglia and Kollias, 2015b). These measurements provide a valuable source of information that can complement surface Doppler measurements, increasing the number of samples in the orbit-to-orbit corrections and further improving their accuracy. Any deviation from the Z-V climatological relationships may indicate a potential antenna mispointing.



## 6 Summary

This study highlights the critical role of precise geolocation and antenna pointing correction in ensuring the quality of EarthCARE's CPR observations, particularly in minimizing errors in Doppler velocity measurements. Through a comprehensive geolocation assessment leveraging natural targets such as coastlines and terrains with significant elevation gradients, we demonstrate that the CPR instrument is properly geolocated within the specified mission requirements. However, the examination of surface Doppler velocity measurements reveals systematic mispointing trends influenced by seasonal variations and thermoelastic distortions of the antenna structure.

The characterization of these mispointing trends, based on surface Doppler velocity measurements, indicates a cyclic pattern in the along-track mispointing angle, which correlates with the spacecraft's daylight cycle. This mispointing is shown to be driven by thermoelastic effects resulting from variations in solar illumination, with peak deviations occurring near daylight entry and a few minutes before exit. The observed biases are quantified and parametrized to a climatological mispointing reference model, which accounts for both phase and amplitude variations throughout the year. This parametrization allows to correct the CPR data to within 5-7 cm/s (the 90th percentile is below 10 cm/s) precision, significantly reducing Doppler velocity biases.

The impact of the antenna mispointing on CPR Doppler velocity measurements is further validated through ice cloud climatology. Prior to correction, the observed reflectivity–Doppler velocity relationships exhibit systematic biases, which are effectively removed after applying the mispointing correction. This improvement confirms that the corrected Doppler velocity data provide a more accurate representation of atmospheric dynamics, ensuring the integrity of EarthCARE's mission objectives.

Overall, the methodologies developed and applied in this study establish a robust framework for geolocation validation and antenna pointing correction that will benefit the ongoing calibration and validation efforts of the EarthCARE mission and will be applied in the level 2 processors; C-APC and C-PRO (Kollias et al., 2023). The findings underscore the necessity of continuous monitoring and refinement of mispointing corrections to maintain the high accuracy required for Doppler velocity measurements, ultimately enhancing the scientific utility of EarthCARE's CPR observations in studying cloud microphysics and precipitation processes.

## Author Contributions

BPT developed the geolocation assessment and antenna pointing correction tools, performed the analysis, generated the figures, and drafted a version of the manuscript. PK, AB, HN and ST contributed to the evaluation of the results and provided feedback on the writing.



**Competing Interests**

None of the authors has any competing interests.

**Financial Support**

Work done by BPT and PK was supported by the European Space Agency (ESA) under the Clouds, Aerosol, Radiation –
Development of INtegrated ALgorithms (CARDINAL) project (grant no. RFQ/3- 17010/20/NL/AD).



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
