# Peer review of "EarthCARE's Cloud Profiling Radar Antenna Pointing Correction using Surface Doppler Measurements"

_EGUsphere, 2025_

## Author Comment (AC2)

**Response to reviewer 2 comments**

The author responses are in blue

**General Assessment**

This paper makes an important contribution to EarthCARE and the achievement of quality Doppler measurements from space. The methodology for addressing satellite pointing accuracy is generally sound, however I have identified several concerns regarding the fundamental assumptions about surface Doppler velocity measurements that should be addressed before publication.

**Major Concerns**

My primary concern stems from the statement on Page 5 Lines 118 that "Therefore any departure from the expected 0 m/s velocity indicates a potential mispointing." I guess this is somewhat true, but the quantitative level of departure will depend not only on the pointing angle of the antenna but also the NRCS of the surface at different angles. The surface backscatter will vary statistically with angle, and this will have the effect of causing non-zero-mean NUBF.

We thank the reviewer for this insightful comment. We agree that surface Doppler velocities can be affected by non-uniform beam filling (NUBF) effects, including those arising from variations in surface backscatter with incidence angle. However, we would like to emphasize that NUBF-induced biases in Doppler velocity are minimal for small angular deviations over spatially homogeneous surfaces (Tanelli et al., 2002). In our case, the NRCS variability within the angular pointing range considered (approximately ±0.01°) is very small. As a result, the NUBF-induced biases over the regions used for the mispointing assessment are negligible at this level of precision.

As noted in the manuscript (line 172), we do apply a correction for NUBF. However, we acknowledge that this correction should be introduced earlier in the text. We have therefore revised the sentence in line 118 to read:

*Therefore, any departure from the expected 0 m/s velocity, after correcting any potential NUBF effects, indicates a potential antenna mispointing.*

There is significant discussion of Figure 3, but very little discussion of Figure 3-a. Figure 3-a appears to show that while the mean surface Doppler velocity of the oceans are consistent with latitude, the Doppler velocity of the land surface varies significantly. This does not appear to be a noise issue, as the standard deviation in Figure 3-b does not show the same features as Figure 3-a. This matches my expectation above.

Having the surface Doppler technique potentially not0 work as well over land does not surprise me, but I'm concerned that if the remainder of this work includes the land surface Doppler velocity, it will cause added uncertainty on the order of 0.5 m/s (a visual estimate of per-latitude mean Doppler changes between land and ocean at the same latitude).

The current text discusses variability (around Line 160) and states that flat surfaces are expected to introduce no vertical motion at nadir. I do not agree with this statement due to how NRCS changes with angle. Further, the data in Figure 3-a do not appear to show a lack of flat-terrain-induced apparent vertical motion. Certain areas of flat land (such as the Great Plains in North America) show significant mean difference from the ocean data, while the Rocky Mountains to their west match the mean ocean velocity.

We thank the reviewer for the observation regarding the differences between land and ocean Doppler velocities in Figure 3a, and the implication that land surface Doppler measurements may introduce additional uncertainties.

Indeed, NUBF effects over land are significantly more pronounced, and there is no well-established methodology for correcting them. The interaction between the radar footprint and the surface, including variable slopes and heterogeneous scattering, introduces uncertainties that are difficult to characterize. Not only orography but surface types also play a key role, affecting both the mean and variability of the Doppler velocity measurements. These effects are often correlated with along-track NRCS gradients. In some regions, the NRCS variability can be up to 30 times higher than that observed over the ocean. For these reasons, land surface Doppler velocity measurements are excluded from the antenna mispointing analysis.

An exception to this is snow-covered land, particularly Antarctica and Greenland. These surfaces are relatively uniform, exhibit low Doppler velocity bias and variability, and are located at latitudes where the CPR operates at high PRF. This leads to improved measurement precision and makes these regions valuable for our antenna mispointing study.

We acknowledge that the description of Figure 3a, the variability discussion and the introduction in section 4 can be expanded for clarity. The revised text now reads:

*While the results highlighted in Figure 3 do not differentiate between ascending and descending orbits, panel (a) already reveals a clear latitudinal structure in surface Doppler velocity over the oceans, suggesting potential mispointing, especially in the northern hemisphere (darker colors). In contrast, land surfaces exhibit considerable spatial variability and regional biases that deviate from the oceanic trend. These biases are not uniformly correlated with orography but are also linked to the heterogenic characteristics of the surface.*

*[…]*

*One of the most notable characteristics of the surface Doppler measurements is their variability, which is dependent to orography, surface type and the CPR PRF settings. The lowest Earth's surface Doppler velocity variability is observed over ocean and snow-covered land (e.g., Antarctica and Greenland). Flat and uniform surfaces are expected to introduce no vertical motion at nadir, whereas heterogenic and rough topography can generate heterogeneous scattering and significant terrain-induced Doppler effects due to slopes and variations in reflectivity causing non-uniform beam filling effects (Manconi et al., 2024).*

*Consequently, land regions tend to exhibit noisier measurements, with exceptions such as the deserts of Western Australia, the Sahara, and Namibia, which have relatively uniform and flat surfaces. Sea ice, on the other hand, appears to considerably increase the measurement variability. Additionally, the high PRF settings, configured to find balance between the unambiguous range and the tropopause height (a proxy for maximum cloud top height) at different latitudes, significantly reduce the measurement variability at high latitudes (e.g., near the North Pole and Antarctica) where the PRF is at its high, further highlighting the influence of the instrument configuration on data quality.*

*[…]*

*The clear-sky Doppler velocity measurements over the ocean (free of ice) and snow-covered land (Antarctica and Greenland) collected for all orbits from June 2024 to February 2025 are used to document the biases observed in the global climatological analysis and, in order to identify any potential antenna mispointing. Other land regions are excluded from the analysis because the high variability of their surface Doppler measurements compromises the precision required for mispointing detection and the integrity of the global assessment.*

The rest of the paper is quite good, but because all the remaining data are effectively zonally averaged, this question of land-vs-ocean remains as a constant source of uncertainty, particularly in the Northern hemisphere.

We would like to clarify that only clear-sky Doppler velocity measurements over the ocean (free of ice) and snow-covered land (Antarctica and Greenland) are used to document the biases observed in the global climatological analysis and to identify potential antenna mispointing (see line 170).

On page 11 Line 255 the paper discusses a technique to ingest 250 km along-track averaged surface Doppler velocity observations. This seems like a good approach (over ocean) but there are no data shown about how this works. The statement around Line 265 that the 90th percentile of residuals remain below 0.00077 degrees is very promising, but is there some data that shows this? I don't see how it can work with the average surface velocity over land varying by ~0.5 m/s as compared to ocean.

We thank the reviewer for highlighting these points. As described in the text, the 250 km window (i.e. about 32 s integration time) is chosen as an optimal balance: it is long enough to smooth out small-scale variability, such as noise, while still preserving meaningful large-scale trends in the Doppler measurements.

Reducing the standard error by a factor of two would require acquiring four times as many observations in the sample, resulting in a window of over two minutes, already too long to capture meaningful mispointing variations at the level of precision targeted in this study.

The data shown in Figure 3 were generated using this 250 km averaging, and the results in Section 4 are both smooth and sufficiently resolved, suggesting that the selected window length is a good choice.

The results referenced in line 265 are based exclusively on surface Doppler velocity measurements over ocean and snow-covered land, using the 250 km along-track averages after correcting for NUBF effects. The 0.00077° value reflects the 90th percentile of residuals between the modeled mispointing trend and the ingested (along-track averaged and NUBF-corrected) values used in the analysis. As previously explained, land measurements (aside from Antarctica and Greenland) are not used in this assessment due to their higher variability and reduced reliability. Therefore, the ~0.5 m/s land-ocean differences do not affect this result.

**Recommendations**

Please address how variations in mean surface Doppler velocity, particularly over land, impact the overall analyses. I recommend performing these same analyses with an ocean-surface mask to determine how land-ocean discrepancies affect the results.

The surface Doppler velocity measurements over land (except for snow-covered regions such as Antarctica and Greenland) are explicitly excluded from the antenna mispointing analysis (line 170). The methodology described in the manuscript is structured to first present the global characteristics of surface Doppler measurements, followed by a careful selection of regions appropriate for a reliable mispointing estimation.

As described in the text, land surfaces are known to introduce additional uncertainties due to NUBF effects and spatial heterogeneity in backscatter and are therefore intentionally omitted from the derivation of the mispointing model.

Introduce land surface Doppler velocity measurements in the mispointing analysis would introduce substantial uncertainties and contaminate the results. Including land data simply to demonstrate that it degrades the results to later apply an ocean-surface mask would obscure the methodology rather than strengthen it.

Please provide a more detailed physical explanation for the observed land-surface Doppler velocity variations.

This has been addressed in previous responses.

**Minor Comments**

Figure 7 - The interpretation of this plot is unclear. It appears to show residual mispointing after removing seasonal effects, but the units are not specified. Please clarify what is being represented.

The figure represents the normalized mispointing trend after removing the seasonal variations. As noted, the mispointing angle is normalized, which is why no physical units are shown. We believe that the current figure caption and corresponding description in the text accurately explain what is being presented. However, if specific elements remain unclear, we would appreciate further guidance from the reviewer on which aspects require clarification.

Line 80  and Figure 1 - Please explain how cross-track geolocation is accomplished with a single overpass using terrain. If this is one overpass it could be informative to show the range-to-surface vs track distance plot combined with the terrain.

Lines 67 to 80 explain how the terrain technique works:

*For significant elevation gradients, the assessment is performed by comparing the instrument's surface detection height to a reference digital elevation model. Artificial mispointing errors are introduced in the along- and cross-track directions, and the absolute geolocation is determined by the shift that maximizes the correlation between the instrument and the DEM-estimated surface height.*

A similar example to what the reviewer suggests, showing range-to-surface versus track distance over terrain is provided in Puigdomènech Treserras and Kollias 2024, Figure 7. Since the focus of the current study is on antenna pointing rather than geolocation, we believe including such a figure here is not essential.

Figure 2 - This figure is challenging to interpret. I recommend:

- Separating the plot into two (ascending and descending) to clarify the different clustering patterns
- Making the stars indicating the mean values more prominent, and
- Replacing the yellow text with a color that provides better contrast.

We thank the reviewer for the helpful suggestions regarding Figure 2. We agree that separating the ascending and descending orbits improves the clarity of the clustering patterns, and we have now split the figure accordingly. In addition, the markers indicating the mean values have been made more prominent, and the coloured texts have been replaced. We believe these changes significantly enhance the figure's interpretability.

[Figure]

Figure 2: Combined global geolocation statistics of the EarthCARE CPR for data collected from August to November 2024: (a) ascending and (b) descending parts of the orbit. Each symbol represents an individual domain where the geolocation is assessed, with the symbol size being indicative of the number of overpasses. Distinctive colors identify different domains, while filled red stars denote the averages. The dashed lines denote the perfect geolocation point (0º).

Conclusion

While the paper represents an important contribution to the field, addressing these concerns - particularly regarding surface Doppler velocity assumptions and land-ocean discrepancies - would significantly strengthen the work.

---

## Author Comment (AC3)

**Response to reviewer 1 comments**

The author responses are in blue.

We thank the reviewer for the positive evaluation of our manuscript and for recognizing the rigor and scientific relevance of our work. We appreciate the reviewer's summary of the study and the remarks regarding the robustness of the methods that will help improve the manuscript completeness and clarity. We have addressed all specific comments below.

**General Assessment**

This study calibrates the antenna pointing of EarthCARE's 94 GHz Cloud Profiling Radar (CPR), the first spaceborne Doppler weather radar, to ensure accurate Doppler velocity measurements. Even small misalignments can bias cloud and precipitation velocities, so the authors focus on two key tasks during the commissioning phase: (1) verifying CPR geolocation and (2) identifying and correcting off-nadir pointing errors using Doppler signals from stationary ground targets. Analyzing surface returns from coastlines, mountains, and snow/ice over the first months of observations (Aug 2024–Feb 2025), they find the CPR is well geolocated (~100 m accuracy) but exhibits subtle, orbit- and season-dependent mispointing. These biases correlate with thermal cycles and are corrected using a climatological mispointing model, reducing velocity errors to within 5–7 cm/s (90% <10 cm/s). Validation using ice cloud data confirms the correction removes spurious Doppler biases. The study concludes that EarthCARE's CPR is now accurately calibrated for high-precision cloud dynamics research.

This work is scientifically rigorous and addresses a vital calibration problem for EarthCARE. The authors thoroughly ground their study in prior literature on sources of Doppler error (spectral broadening, non-uniform beam filling, and pointing uncertainty) and build on pre-launch plans for EarthCARE's calibration (citing Kollias et al. 2023 for broadening/NUBF corrections and earlier studies like Tanelli et al. 2005 for pointing issues). The methods used are appropriate and appear very robust. The strategy of using Earth's surface as a calibration target is sound: a stationary ground return should have zero Doppler shift (aside from known platform motion components), so any systematic offset directly indicates a pointing error. A potential weakness in the methodology is that some choices and corrections are referenced to other documents and could be explained in more detail for completeness. For instance, the geolocation technique could be explained a little bit more. In general the article is well written, it requires mostly minor corrections, and I find just one major issue:

1. The conclusion that thermoelastic deformation from solar heating causes the mispointing is based on circumstantial evidence (correlation with day/night cycle and seasonal repetition). The authors have made a strong case for it, but direct evidence (e.g. temperature measurements on the radar structure) was not presented. EarthCARE likely has temperature sensors on the CPR or nearby structure. A correlation between the measured antenna/baseplate temperatures and the inferred pointing bias could conclusively link cause and effect

The limited publicly available information regarding the pre-launch ground testing of the CPR in a simulated space environment supports the reviewer's comment about "…circumstantial evidence…" in this study. In other words, the reviewer is correct to point out this major issue. However, we would like to clarify a few things. First, satellite antennas are well known to experience thermal deformation in the alternating hot and cold space environment. There are a lot of examples in literature where such effects have been observed in orbit. JAXA did conduct pre-launch tests to evaluate the level and behavior of the thermal deformation of the CPR 2.5 m diameter CPR reflector in a simulated space environment. Several thermistors were placed in the back of the CPR reflector and detailed measurements of the surface deformation were performed. JAXA did analyze the pre-launch measurements and verified that the CPR reflector antenna will undergo significant thermal deformation in-orbit that should be corrected to produce unbiased Doppler velocity estimates. JAXA did parameterize the thermal deformation of the CPR antenna using the set of temperature measurements and the correction was applied during the early phase of the commissioning phase.

However, the CPR engineers and scientists realized that the correction was introducing artifacts that were not consistent with the results acquired using the Earth's surface as a calibration target. Currently, the initial parameterization of the CPR antenna deformation using the set of temperature measurements in the back of the CPR reflector has been removed and the Earth's surface is the main method to correct the CPR antenna pointing. Soon, JAXA plans to present a new parameterization for the CPR antenna pointing that will use the temperature measurements. The authors do not have access to these temperature measurements; thus, it is difficult to demonstrate their relevance and potential to correct for the CPR antenna pointing.

**Minor Points**

1. L67 – Methodology for Mispointing Detection in Areas with Large Elevation Gradients:

Please clarify the methodology used to detect mispointing in regions with complex topography. Specifically:

- What are the "artificial mispointing errors" referred to here?

We thank the reviewer for this request for clarification. The "artificial mispointing errors" refer to deliberate shifts that we introduce in the along- and cross-track coordinates of the instrument's geolocation when projecting the detected surface height onto the reference digital elevation model (DEM). Specifically, we apply small incremental angular offsets in both directions. The step size is chosen such that the corresponding horizontal displacement at the surface matches the resolution of the DEM (1 arc-second), ensuring optimal sampling for the correlation analysis.

- Are these based on simulations of surface returns assuming perfect satellite geolocation, where only antenna azimuth and elevation are varied and then compared to the actual radar signal?

No, the artificial mispointing errors are not based on simulations of surface returns assuming perfect satellite geolocation. Instead, they are implemented as systematic angular shifts applied to the CPR data geolocation coordinates. We use real CPR surface detection measurements and then artificially vary the assumed along- and cross-track pointing angles used to project those detections onto the DEM. For each trial pointing angle, we compute the correlation between the CPR and DEM-estimated surface height. The pointing angle that yields the highest correlation is considered the best estimate of the actual geolocation offset. Therefore, this method relies on the analysis of real measurements, not synthetic simulations.

- If so, please describe this process more explicitly, including assumptions and limitations. For the coastline analysis, explicitly state that the land and ocean have distinct radar backscatter signatures ($\sigma_0$), which allows the land–ocean transition to be used for detecting pointing biases.

Here is a revision of the paragraph addressing the reviewer's concerns more explicitly (L65):

Here, the geolocation assessment is performed using the techniques described in Puigdomènech Treserras and Kollias 2024, based on the positions of known natural targets, such as significant elevation gradients and coastlines over more than 140 domains of 2x2 degrees distributed around the globe.

For significant elevation gradients, the assessment is performed by comparing the instrument's surface detection height to a reference digital elevation model (DEM). To do this, small displacements are systematically applied to the CPR geolocation coordinates in both along- and cross-track directions when projecting the detected surface onto the DEM. These displacements correspond to different possible geolocation offsets, as any pointing results in a lateral shift of the projected footprint on the ground. The step sizes are chosen such that the corresponding horizontal shifts match the DEM resolution (1 arc-second), ensuring optimal sampling for the analysis. The absolute geolocation is determined by the shift that maximizes the correlation between the instrument and the DEM-estimated surface height.

For coastlines, the analysis leverages the fact that land and ocean surfaces exhibit distinct normalized radar cross section signatures, resulting in sharp surface signal gradients at land-ocean transitions. These transitions, detected in the CPR surface signal, provide coastline geolocation markers. Then, through a minimization approach, the absolute geolocation is determined by minimizing the error between a collection of coastline detections and a reference map. The primary limitation of this approach is that it requires sufficient sampling of coastline crossings to ensure statistical robustness, which is why detections over several months must be aggregated.

2. Figure 1 – Description and Interpretation

We thank the reviewer for these helpful suggestions to clarify Figure 1, which was intended to illustrate examples of the geolocation analysis performed using both mountainous and coastline regions.

This figure needs a more detailed explanation:

- Does panel **b** represent the optimal mispointing correction for the entire domain shown in panel **a**, or is it specific to a selected location along the track?

Panel (b) represents the correlation analysis and optimal geolocation offset (expressed as a shift in the along- and cross-track directions) estimated for the entire 2x2 domain shown in panel (a), using all CPR surface detections within that domain. The analysis is not performed at a single location along the track, but rather over the full set of surface detections in the domain during the overpass.

- Clarify how both panels relate to the region over the Greek Islands.

Panels (a) and (b) correspond to the mountainous region in British Columbia, Canada (not the Greek Islands). The Greek coastline case is shown in panel (c), which is derived from an independent analysis using coastline detections collected over several months (August to November 2024). Panel (c) illustrates the spatial distribution of the observed coastline transitions used in the geolocation assessment for that region.

- Additionally, please provide an equivalent of panel **b** using the coastline detection method, for direct comparison between methods.

We appreciate this suggestion. In principle, one could construct a 2D representation of the coastline alignment error as a function of along- and cross-track offsets, analogous to the correlation figure shown in panel (b). However, the coastline analysis is based on directly minimizing the spatial distances between detected coastline transitions and the reference coastline map, using a minimization rather than an exhaustive scan over a grid of possible offsets. As a result, the analysis does not naturally yield a 2D figure comparable to that shown in panel (b). Additionally, since coastline crossings are sparse and irregularly distributed, the resulting cost function does not provide a smooth, high-resolution 2D structure comparable to that obtained with elevation gradients. For these reasons, we chose to present the coastline results in panel (c) as the spatial distribution of the detections used in the minimization, which more directly illustrates the nature of the data and the method employed. We will clarify this point in the revised figure caption:

Figure 1: EarthCARE CPR geolocation assessment using significant elevation gradients and coastlines. Panels (a) and (b) illustrate an example based on significant elevation gradients in a mountainous region of British Columbia, Canada. Panel (a) shows the selected 2x2 domain, with the red line representing one of the EarthCARE overpas. Panel (b) depicts the correlation analysis used to estimate the optimal geolocation offset for the full set of surface detections and entire domain shown in panel (a), with the white line representing the satellite path, in descending orbit, and the red filled star denoting the final estimated geolocation offset. Panel (c) presents the coastline-based geolocation assessment in a Greek Islands region. The red dots represent clear coastline detections, aggregated from multiple overpasses between August and November 2024. Unlike the elevation-gradient method, the coastline analysis is based on direct

minimization of spatial distances between the detected transitions and the reference coastline map, rather than a 2D scan over a grid of possible offsets. The base map is © OpenStreetMap contributors 2015, distributed under the Open Data Commons Open Database License (ODbL) v1.0.

3. L80+ – Time-Varying Pointing Correction

You show that antenna mispointing varies over time. This temporal evolution challenges the coastline-based detection method, which requires several months of data to achieve sufficient spatial sampling. Please discuss this limitation more clearly and consider quantifying the error introduced when using long-averaged coastline data under varying pointing conditions.

We thank the reviewer for this important point. It is indeed correct that the time-varying nature of the antenna mispointing introduces a limitation for the coastline-based geolocation assessment, which relies on aggregating several months of coastline detections to achieve sufficient spatial sampling. We have clarified this limitation in the manuscript (see the revised text in the answer to minor point 1, above).

Regarding the potential error introduced by using long-averaged coastline data under varying pointing conditions: while the antenna mispointing does evolve with season and solar illumination conditions, its variation over the typical few-month period used in the coastline analysis is relatively small in terms of its impact on geolocation. We believe that it is important to distinguish between geolocation and antenna mispointing errors, as they affect the measurements at different scales. The resulting impact of aggregating data over several months on the coastline-based geolocation estimates is expected to remain stable. The following clarification will be added to the text:

While the effect of aggregating data over several months could in principle smooth out geolocation variations, the technique is based on the assumption that such variations are sufficiently small, and the resulting estimates are expected to remain stable over timescales of a few months.

4. L116: Spell out "100s" as "hundreds" for clarity

Done. Thanks.

5. L120: Mention explicitly that the surface Doppler velocity analysis is performed globally, without separating land and ocean scenes.

Done. Thanks.

6. L131: Clarify what is meant by "surface Doppler velocity."

- Is it the Doppler velocity at the radar signal peak, or a mean over a defined range around the peak?
- Given that the CPR's point target response is broad and flat, explain how the surface location is selected in the Doppler spectrum and how consistent this is across scenes.

We thank the reviewer for this useful request for clarification. In this study, surface Doppler velocity refers to the Doppler velocity value assigned to the range bin corresponding to the detected surface. The surface detection is part of the L1b algorithms and is based on a parabolic fitting of the reflectivity profile near the surface. This approach allows the surface location to be estimated with sub-bin precision (bin number and fractional bin).

In our analysis, we use the Doppler velocity corresponding to the integer range bin reported as the detected surface bin by the L1b product. This provides a consistent and robust definition of surface Doppler velocity across different locations.

We have included this clarification in the text at L132:

The spatial average and variability of the surface Doppler velocity (i.e. the Doppler velocity corresponding to the surface range) for the period from August to November 2024 is shown in Figure 3a,b. The surface range is identified in the CPR L1b surface detection algorithm, which applies a parabolic fitting of the reflectivity profile near the surface. Here, the surface Doppler velocity corresponds to the Doppler velocity at the integer range bin reported by this detection.

7. Provide Parametrization in Usable Form:

- Please provide the Fourier expansion of the normalized temporal trend (ranging from -1 to 1) for a reference day (e.g., January 1).
- Include the same expansion for the minimum, maximum values and temporal shift.
- Express the correction directly in terms of Doppler velocity, not in antenna angle, so users can directly apply it to Level 1 data without relying on Level 2 products.

We thank the reviewer for this suggestion. While we fully agree on the importance of providing a practical correction method for users, we caution that parametrizations do not capture well the trends presented in this study and are therefore not recommended. The phase shift of the antenna mispointing (linked to the spacecraft's daylight entry and exit times) does follow a smooth seasonal variation that can be modelled with a harmonic function. However, there is no evidence that the amplitude variations follow a simple harmonic pattern and, after analyzing more data, we've found that they actually exhibit irregular behavior that cannot be adequately represented with a Fourier expansion without introducing significant residual errors. Other attempts to parametrize the normalized trend, including polynomials of more than 10 degrees, also tend to introduce residual errors, particularly near the transitions and extremes of the trend.

Regarding the request to express the correction directly in Doppler velocity terms: the correction must be applied in covariance space to properly avoid aliasing effects. Applying the correction directly in Doppler velocity space would lead to ambiguities and is therefore not recommended.

For all these reasons, and to properly address the reviewer suggestions, we decided to provide a correction in the form of a Look-Up Table (LUT), which we include in the Data Availability section. This approach will allow us to update the LUT as more data becomes available or in case of orbital modifications that could affect the mispointing pattern. Additionally, we have included an ANNEX that describes how to utilize the LUT information to correct the L1 Doppler velocity data. The text is provided at the end of the comments.

8. Section 4 – Comparison with High-Gradient Land Surface Method

- Include a marker (e.g., star/square/dot) on the correction plots to indicate results from the mispointing estimates derived over topographically complex land areas (as described earlier).
- Assess how these compare with the Doppler-based correction estimates.
- Use consistent marker colours for the same observation week across methods to visually indicate agreement or differences.

We thank the reviewer for this suggestion. However, we respectfully note that such a comparison is beyond the intended scope of this paper. The focus of the manuscript is the characterization and correction of antenna mispointing, as inferred from surface Doppler velocity measurements. The geolocation assessment presented in Section 2 is included to demonstrate the overall geolocation accuracy of the CPR instrument, which is of relevance here, but is not intended to serve as an alternative or reference method for quantifying antenna mispointing.

As previously explained, it is important to distinguish between geolocation and antenna mispointing errors, as they affect the measurements at different scales. The antenna mispointing trend presented in this study

has an amplitude span of approximately 0.006º. At a satellite altitude of 395 km, a mispointing error of 0.01º corresponds to a geolocation error of approximately 69 m, while the high-resolution DEM used in the geolocation analysis has a spatial resolution of 1 arc-second (~30 meters). The sensitivity and resolution of the geolocation method are not sufficient to resolve the small variations in antenna mispointing characterized in Figure 4.

For these reasons, we believe that adding such markers or performing a direct visual comparison in Figure 4 would not provide meaningful additional insight, and would risk introducing misleading interpretations or obscuring the trends that the figure is intended to highlight.

9. Section 5: In Figure 8, please add a line showing the global V–Z (Doppler velocity–reflectivity) relationship derived from all valid cirrus cloud observations (i.e., those not affected by the demodulation bias), not just January data. Alternatively, provide a supplementary figure showing the V–Z relationship for cirrus clouds, including the standard deviation envelope for context. Use all the data together over all frames and provide a polynomial fit to the formula.

We thank the reviewer for this suggestion. However, we respectfully note that providing a global V–Z parametrization is beyond the intended scope of this paper. The goal of Section 5 is to illustrate the impact of antenna mispointing correction on Doppler velocity measurements, using cirrus cloud observations as an independent validation case. The month of January was specifically selected to highlight these effects clearly, as extending the analysis to a multi-month dataset would smooth out the mispointing signal and dilute the demonstration.

Moreover, global Z–V relationships are known to vary significantly with temperature. Providing such a parametrization here, without detailed analysis of these dependencies, could risk introducing confusion or misleading interpretations.

A more detailed analysis of the impacts of antenna pointing characterization, has been performed and is presented in Kim et al. (2025). We therefore prefer to maintain the current scope of this section but we will add such reference to the text.

================================================================================

**ANNEX I**

This section describes the application of the antenna mispointing Look-Up Table (LUT) to correct the EarthCARE CPR Level 1b Doppler velocity data using the climatological fit of the CPR antenna mispointing presented in this study. The correction is applied directly to the complex lag-1 autocovariance of the pulse-pair radar signal, prior to Doppler velocity derivation.

Performing the correction directly in Doppler velocity space requires careful handling of Nyquist folding effects, particularly at high PRF. Even small mispointing angles can induce phase shifts that exceed the Nyquist limit, leading to velocity aliasing. Instead, applying the correction at the level of the complex radar signal avoids this ambiguity and ensures phase continuity.

**Overview**

The EarthCARE CPR Doppler velocity is derived from the phase angle of the lag-1 autocovariance of the IQ signal. This phase shift between consecutive pulses encodes the Doppler frequency and is given by:

$$V = \frac{\lambda \cdot \mathrm{PRF}}{4\pi} \cdot \theta_{nom},$$
(A1)

Where $\lambda$ is the radar wavelength, $\mathrm{PRF}$ is the pulse repetition frequency and $\theta_{nom}$ is the phase angle of the complex lag-1 autocovariance, computed from its real ($R[R]$) and imaginary ($I[R]$) components:

$$\theta_{nom} = \mathrm{atan}(I[R], R[R]),$$
(A2)

**Line-of-sight correction**

A correction must first be applied for line-of-sight (LOS) contamination resulting from the satellite's motion projected onto the CPR beam direction. This effect is not accounted for in the pulse-pair radar signal reported in the L1b. The LOS-projected velocity ($V_{LOS}$) is computed as follows:

$$V_{LOS} = V_{\text{sat}} \cdot \sin(\theta_{ADS}), \tag{A3}$$

Where $V_{\text{sat}}$ is the satellite velocity in Earth-Centered, Earth-Fixed (ECEF) coordinates and $\theta_{ADS}$ is the antenna pitch angle reported by the Attitude Determination System (ADS). This LOS velocity introduces a phase shift in the measured signal:

$$\theta_{LOS} = \frac{4\pi \cdot V_{LOS}}{\lambda \cdot \text{PRF}}, \tag{A4}$$

The phase correction is applied by subtracting this LOS-induced phase ($\theta_{LOS}$) from the nominal measured phase:

$$\theta_{LOS-CORR} = \theta_{nom} - \theta_{LOS}, \tag{A5}$$

To correct the complex lag-1 autocovariance, the real and imaginary components of the lag-1 autocovariance are recomputed using the corrected phase:

$$I[R]_{LOS-CORR} = |R[R] + j \cdot I[R]| \cdot \sin(\theta_{LOS-CORR}), \tag{A6}$$

$$R[R]_{LOS-CORR} = |R[R] + j \cdot I[R]| \cdot \cos(\theta_{LOS-CORR}), \tag{A7}$$

The Doppler velocity corrected for LOS contamination can then be obtained by applying equations (A2) and (A1) to the updated complex radar signal defined by (A6) and (A7).

**Antenna mispointing correction**

The antenna mispointing Look-Up Table (LUT) provides a normalized mispointing pattern as a function of ANX time, along with the corresponding seasonal amplitude and phase shifts. These parameters define a climatological model of the antenna mispointing that evolves smoothly over the course of the year. At a given ANX time (t) and day-of-year (d), the mispointing correction is computed as follows:

$$\theta_{APC}(t, d) = m_{\text{norm}}\left(t + \delta t_\phi(d)\right) \cdot \left(A_{max}(d) - A_{min}(d)\right) + A_{min}(d), \tag{A8}$$

Where $m_{\text{norm}}(t)$ is the normalized mispointing pattern, $\delta t_\phi(d)$ is the seasonal phase shift and $A_{min}(d)$ and $A_{max}(d)$ are the minimum and maximum seasonal amplitude bounds.

This parametrization allows the reconstruction and correction of the antenna mispointing angle across the orbit and throughout the year. Once the mispointing angle $\theta_{APC}$ is known, it can be converted to a Doppler phase correction following the same approach described in the LOS correction section.

**Implementation notes**

The LUT information must be applied to each specific frame by interpolation. All required variables are found in the CPR L1b data product (C-NOM), including *rayHeaderLambda* ($\lambda$), *rayStatusPrf* (PRF), *covarianceCoeff* ($R[R]$ and $I[R]$), *pitchAngle* ($\theta_{ADS}$), *satelliteVelocityX*, *satelliteVelocityY*, *satelliteVelocityZ* (components of $V_{\text{sat}}$), *profileTime* and *ANXTime* (t).

In the CPR L2a processing (C-APC), an additional optimization step is applied to minimize the residuals between the climatological mispointing model and the mispointing angles inferred from the raw measured surface Doppler velocity measurements. This step refines the amplitude scaling for each orbit and ensures residual Doppler velocity biases are reduced below 10 cm/s in most cases.

This correction is valid at the time of reviewing this manuscript (May 2025). Future versions of the CPR L1b data product may include antenna mispointing corrections directly in the processing chain. Additionally,

updates to the orbital specifications may affect the accuracy of this correction. Users are advised to consult the latest product specifications and orbital parameters before applying this method.